# Cytochalasins Suppress 3D Migration of ECM-Embedded Tumoroids at Non-Toxic Concentrations

**DOI:** 10.3390/ijms26147021

**Published:** 2025-07-21

**Authors:** Klara Beslmüller, Lieke J. A. van Megen, Timo Struik, Daisy Batenburg, Elsa Neubert, Tom M. J. Evers, Alireza Mashaghi, Erik H. J. Danen

**Affiliations:** Leiden Academic Centre for Drug Research, Faculty of Science, Leiden University, 2333 CC Leiden, The Netherlands; k.beslmuller@lacdr.leidenuniv.nl (K.B.); l.j.a.van.megen@umail.leidenuniv.nl (L.J.A.v.M.); timostruik1997@gmail.com (T.S.); d.batenburg@lacdr.leidenuniv.nl (D.B.); e.neubert@lacdr.leidenuniv.nl (E.N.); t.evers@amolf.nl (T.M.J.E.); a.mashaghi.tabari@lacdr.leidenuniv.nl (A.M.)

**Keywords:** tumoroid, stiffness, cofilin, F-actin dynamics, cancer cell migration, extracellular matrix (ECM), cytochalasins, migrastatic

## Abstract

Migrastatic strategies are considered as candidate therapeutic approaches to suppress cancer invasion into local surrounding tissues and metastatic spread. The F-actin cytoskeleton is responsible for key properties regulating (cancer) cell migration. The cortical F-actin network controls cell stiffness, which, in turn, determines cell migration strategies and efficiency. Moreover, the dynamic remodeling of F-actin networks mediating filopodia, lamellipodia, and F-actin stress fibers is crucial for cell migration. Here, we have used a conditional knockout approach to delete cofilin, an F-actin-binding protein that controls severing. We find that the deletion of cofilin prevents the migration of cancer cells from tumoroids into the surrounding extracellular matrix without affecting their viability. This identifies cofilin as a candidate target to suppress metastatic spread. Pharmacological inhibitors interfering with F-actin dynamics have been developed but their effects are pleiotropic, including severe toxicity, and their impact on 3D tumor cell migration has not been tested or separated from this toxicity. Using concentration ranges of a panel of inhibitors, we select cytochalasins based on the suppression of 2D migration at non-toxic concentrations. We then show that these attenuate the escape of tumor cells from tumoroids and their migration into the surrounding extracellular matrix without toxicity in 3D cultures. This effect is accompanied by suppression of cell stiffness at such non-toxic concentrations, as measured by acoustic force spectroscopy. These findings identify cytochalasins B and D as candidate migrastatic drugs to suppress metastatic spread.

## 1. Introduction

Cancer cells are often softer than healthy cells [1,2] whereas tumors themselves are typically stiffer than healthy tissue. The cortical actin network, composed of a thin network of F-actin filaments and actin-binding proteins aligning the plasma membrane, is the most important determinant of cell stiffness [3]. Stiffness is modulated by cytoskeletal dynamics involving the assembly and disassembly of actin fibers under the control of a variety of actin-binding proteins, myosin-driven contraction of the actin cytoskeletal fiber network, and cell adhesion complexes [4,5]. One key regulator of the actin cytoskeleton is cofilin-1 (CFL). It binds F-actin filaments to weaken the packing of g-actin subunits causing filament severing, which, in turn, also provides new barbed ends for new F-actin branches [6]. Reduced CFL activity has been associated with stiffening of endothelial cells [7], but depletion of CFL has been shown to cause a decrease in contractility and reduced cortical stiffness in melanoma cells [8]. Thus, the assembly and disassembly of F-actin networks, such as the cortical actin network controlling cell stiffness, are tightly regulated by actin-binding proteins, including CFL [9], but their role in cell migration, which requires extensive actin remodeling, is incompletely understood.

Several inhibitors have been developed that interfere with cellular actin dynamics (Figure 1) [10,11,12,13]. These include drugs such as Jasplakinolide that stimulate F-actin polymerization and bind and stabilize F-actin filaments and networks. By contrast, drugs such as Latrunculin A bind g-actin monomers and interfere with F-actin filament formation. Cytochalasins (e.g., cytochalasin B (cyB) and (cyD)) are mycogenic toxins that also reduce actin polymerization, in this case by binding to F-actin barbed ends [14]. CyB slows down g-actin incorporation at the barbed end [15] and inhibits filament–filament interactions, thus suppressing network formation [11]. CyD is more potent in suppressing F-actin filament formation; it stimulates g-actin dimer formation and subsequent ATP hydrolysis, leading to the dissociation of actin monomers [16]. While its effect is primarily through a direct interaction with F-actin, cyD also interferes with CFL–actin interactions and regulates CFL phosphorylation, which provides an additional indirect role for cyD in the regulation of actin dynamics [17,18].

By disturbing the formation and dynamics of the various F-actin filamentous networks, cytochalasins interfere with key cellular functions, including cell division, adhesion, and migration [19]. As cytochalasins disrupt the cortical actin network, they are expected to decrease cellular stiffness. Indeed, cyB and cyD have been observed to cause the softening of different types of cancer cells [20,21,22]. Cellular softening and interference with filamentous actin networks driving lamellipodia and other promigratory structures are expected to interfere with cell migration. For cyD, the inhibition of the 2D random migration of breast and lung cancer cells has been reported at a concentration of 0.1 µM [23], and the inhibition of the migration of melanoma cells through a collagen gel was observed at 50 nM [24]. While the impact of cytochalasins on epithelial-to-mesenchymal transitions and the growth of tumoroids has been reported [25,26], their effect on migration of cancer cells in the context of 3D tumor tissue is unresolved. Moreover, cytochalasins have so far been studied at concentrations expected to cause toxicity, precluding longer-term monitoring of their effects on cell migration.

Taken together, interfering with the actin cytoskeleton has pleiotropic effects, including an impact on cell survival, cell division, cortical stiffness, and cell migration. To investigate these effects in the context of cancer, we employed a conditional knockout model and pharmacological inhibitors to target the migration of cancer cells from tumoroids into the extracellular matrix (ECM). We show that the deletion of CFL interferes with the migration of cancer cells from tumoroids into the surrounding ECM without affecting their viability. We find that out of several inhibitors tested, only cytochalasins inhibit the escape of tumor cells from tumoroids and their migration into the surrounding ECM at non-toxic concentrations, and this is accompanied by the suppression of cell stiffness. These findings point to CFL as a candidate target and cyB and cyD as candidate migrastatic drugs to suppress metastatic spread.

## 2. Results

### 2.1. Deletion of CFL Attenuates Migration of Cancer Cells from ECM-Embedded Tumoroids

We generated conditional knockout models for CFL in Hs578T and MV3 cells using two sgRNAs for each gene. In Hs578T cells, each sgRNA was tested separately. As the knockout procedure was inefficient in MV3, a combination of both sgRNAs was used in these cells for a successful knockout population. The cells were treated with doxycycline (dox) for 4 days (Hs578T) or for 3 days (MV3) and then left in medium without dox for a minimum of 2 days before the start of the measurement. Dox treatment caused the depletion of CFL in the Hs578T and MV3 cells (Figure A1). The depletion of CFL slightly reduced the 2D growth rates of the Hs578T and MV3 cells (Figure 2A).

In agreement with its role in F-actin severing, the depletion of CFL in each cell line increased the accumulation of F-actin at the cell cortex (Figure 2B; note that throughout this paper the F-actin cytoskeleton is visualized in green for Hs578T and in red for MV3). Consistent changes in cell shapes or sizes were not observed in any of the knockouts. Next, we determined how the depletion of CFL affected cell migration from tumoroids into the surrounding ECM scaffold. To control for effects on cell viability in this setup, the tumoroids were stained with Propidium Iodine (PI) for live confocal imaging and the detection of dead cells, prior to fixation and F-actin staining. Treatment with 20% ethanol was used to induce cell death as a positive control for PI staining. These experiments showed that the depletion of CFL did not induce cell death in either of the cell lines (Figure A2). To quantify 3D migration, we calculated the area of the z-projection of all imaged z-slices. For the Hs578T tumoroids, CFL knockout effectively reduced the migration of cells escaping from the tumoroid and moving into the collagen (Figure 3A,B). The protein tyrosine kinase inhibitor bosutinib [27] was used as a positive control in these experiments, which suppressed 3D cell migration without affecting growth at a concentration of 1.25 µM (Figure A3). For MV3, a similar, albeit less strong, inhibition was observed. Together, these experiments show that the depletion of CFL suppresses the ability of tumor cells to escape from a tumoroid and travel into the surrounding ECM, without affecting cell viability.

### 2.2. At Non-Toxic Concentrations, Jasplakinolide, Latruncilin A, Blebbistatin, and GSK-269962 Do Not Affect 2D Migration Speed

Following the genetic targeting of the F-actin regulator CFL, we employed pharmacological strategies to target actin cytoskeletal dynamics (Figure 1). While the selected compounds have been extensively used in various experimental setups, it remains unclear whether they exert migrastatic effects that are independent of cytotoxicity, nor have they been tested for suppressing tumor cell migration in 3D ECM environments. In addition to compounds targeting F-actin turnover, with a mode of action as was discussed in the introduction section, we suppressed actomyosin contractility using GSK-269962 targeting ROCK [28], thus inhibiting myosin light-chain phosphorylation [29,30], and the myosin II inhibitor blebbistatin [31].

To determine the highest non-toxic concentrations, the inhibitors were tested on Hs578T cells for their effect on proliferation. The exposure of 2D cell cultures to concentration ranges of the inhibitors for 24 h showed that Jasplakinolide was well tolerated up to ~100 nM, but at the higher concentrations tested, cell growth showed a rapid decline (Figure 4, left panels). Exposure to Latrunculin A showed a more gradual growth suppression beyond ~37 nM. Blebbistatin caused a sharp drop in cell growth beyond ~37 µM, and GSK-269962 caused a gradual decline at concentrations ranging between 100 nM and 1 µM and a strong growth suppression at 3 µM. At these concentrations the inhibitors also triggered changes in the F-actin cytoskeletal architecture, with a loss of stress fibers and the appearance of ruffle-like structures (Figure A4).

The effect of non-toxic concentrations of these inhibitors on 2D random cell migration was tested by the real-time imaging of subconfluent cultures exposed to similar concentration ranges of the inhibitors. At concentrations below the initiation of growth suppression, none of these inhibitors suppressed cell migration speed in this setup (Figure 4, right panels). Bosutinib was used to suppress 2D cell migration without affecting growth at a concentration of 1.25 µM (Figure A3). The cells were fixed and stained to visualize the F-actin cytoskeleton. At non-toxic concentrations, Jasplakinolide did not markedly affect F-actin morphology, whereas treatment with Latrunculin A, blebbistatin, or GSK-269962 led to reduced F-actin stress fibers and the enhanced formation of membrane ruffles (Figure A4).

### 2.3. CyB and cyD Suppress 2D Migration Speed and Cell Stiffness at Non-Toxic Concentrations

We tested two cytochalasins, cyB and cyD, in the same manner. Exposure to cyB concentrations just below the initiation of growth suppression at ~750 nM led to a reduction in 2D random migration speed in Hs578T cells (Figure 5A,B). A similar trend was observed when MV3 human melanoma cells were exposed to concentrations up to 247 nM cyB, beyond which growth suppression became obvious in this cell line, but here the suppression of migration did not reach significance. In Hs578T cells, F-actin stress fibers gradually disappeared with increasing concentrations of cyB as actin aggregates and structures resembling membrane ruffles appeared (Figure 5C,D). For MV3 cells similar effects were observed, but these became apparent more abruptly when passing from ~750 nM to the µM dose range. Polyploid cells appeared at high concentrations of cyB for both cell lines, and MV3 showed extremely large flattened multinucleated cells at 2 µM.

We then assessed the effect of cyB and cyD on cell stiffness using AFS. Cells were plated together with silica beads inside a chip and a constant acoustic force was applied, pushing the bead upwards and thereby stretching the cells, with the amount of stretch inversely related to stiffness (Figure 5E). AFS measurements showed that cyB, at concentrations not affecting cell growth but already suppressing 2D random cell migration, strongly reduced cell stiffness. Cells exposed to the control medium became stiffer over a 24 h time period in the chip (134% for Hs578T; 1640% for MV3). Exposure to a medium containing cyB reduced such stiffening, resulting in an elastic modulus of 44% and 114% of the elastic modulus measured on the chip before treatment for Hs578T and MV3, respectively (Figure 5F).

While Hs578T and MV3 showed comparable sensitivities to cyB (Figure 5A), cyD was much less cytotoxic in MV3 cells, where up to ~1 µM was well tolerated, as compared to Hs578T, where ~100 nM cyD almost completely blocked growth (Figure 6A). In Hs578T, no inhibition of 2D random migration was observed at non-toxic concentrations, but cyD suppressed MV3 migration at 0.4 and 1 µM without affecting 2D growth (Figure 6B). The effects of cyD on F-actin cytoskeletal morphology appeared similar to what was observed for cyB (Figure 6C,D). Likewise, AFS experiments showed that, similar to cyB, non-toxic concentrations of cyD markedly suppressed the cortical stiffness of Hs578T and MV3. The stiffening in the chip of Hs578T cells dropped from 134% to 29% with 41 nM cyD (Figure 6E). For MV3 cells this difference was 1640% versus 33%.

### 2.4. Cytochalasins Are Migrastatic in the Context of 3D ECM-Embedded Tumoroids

Having observed that cyB and cyD suppress cell stiffness and somewhat reduce 2D random cell migration at concentrations below those causing growth suppression, we next asked if these inhibitors may suppress the traveling of tumor cells from a tumor into a surrounding 3D ECM. For this, 3D ECM-embedded tumoroids were generated. The translation of the results obtained for 2D migration to cells navigating a 3D ECM network is not straightforward as cells employ highly distinct mechanisms under these conditions. Nevertheless, as a preselection for optimal concentrations this approach worked well. We found that the concentrations of cyB and cyD selected in the 2D setup could be translated to the 3D ECM-embedded tumoroids, and in 3D their migrastatic effects were more pronounced. At the concentrations tested, no signs of cell death marked by PI staining were observed for Hs578T or MV3 tumoroids exposed to cyB or cyD (Figure 7A and Figure 8A). After live cell imaging, the tumoroids were fixed and stained with phalloidin, and the migration of tumor cells into the ECM was analyzed. Both the Hs578T and MV3 tumoroids showed extensive migration, and for both tumors cyB and cyD attenuated ECM penetration in a concentration-dependent manner across the range of non-toxic concentrations tested (Figure 7B,C and Figure 8B,C). Lastly, the highest concentrations of cyB and cyD inhibiting migration without cytotoxicity in the Hs578T and MV3 tumoroids were tested on 3D ECM-embedded hTERT-immortalized human fibroblasts. Background PI staining was observed in these cultures, and cyB caused a slight increase while cyD had no effect (Figure 8D). The positive control ethanol strongly increased PI staining. Moreover, brightfield imaging showed that clusters remained intact in the presence of cyB or cyD in contrast to ethanol, which showed a cloud of small dead cells, indicating that in this 24 h exposure assay, the migrastatic concentrations of cyB and cyD were tolerated by fibroblasts.

Together, these experiments identify cyB and cyD as migrastatic compounds with a favorable efficacy/cytotoxicity profile in a 3D microtissue setup, thus representing candidate inhibitors for attenuating tumor invasion of surrounding tissues.

## 3. Discussion

Migrastatic drugs have been proposed to suppress cancer invasion into tissues surrounding a primary tumor and into distant organs penetrated by metastasizing cells [32]. As actin dynamics are key to most, if not all forms of cell migration, interfering with this process may represent such a migrastatic strategy. However, given the prominent role of the F-actin cytoskeleton in many cellular processes, including those affecting cell survival, toxicity is a major obstacle. Here, we have (1) used a genetic approach to identify CFL as a candidate target for migrastatic strategies, (2) tested for drugs affecting actin dynamics and contractility as well as the impact of concentration ranges on cell survival, stiffness, and migration in 2D and 3D models to identify cyB and cyD as candidate migrastatic drugs that suppress cancer cell migration at concentrations not causing reduced cell viability.

Making use of conditional knockouts for CFL, we find that CFL supports tumor cell stiffness. There are earlier reports indicating that loss of CFL activity can lead to either increased or decreased cell stiffness [7,8]. In agreement with the severing function of CFL, increased F-actin staining by Phalloidin was observed by us in CFL knockout cells, but cell shapes were not affected. Nevertheless, CFL depletion strongly reduced the migration of tumor cells in 3D ECM scaffolds. Others have shown that loss of CFL leads to increased stress fiber formation and interferes with dynamic protrusions [33]. By localizing cofilin’s activity, tumor cells tightly regulate where actin remodeling occurs, leading to the efficient and optimized formation and movement of invadopodia and lamellipodia [34]. Depleting the full severing activity of ADF/CFL leads to a loss of cell viability [33], but we find that the deletion of CFL alone does not affect viability. The pathway regulating CFL activity has been proposed to present a candidate target in cancer [35], and our findings indicate that targeting CFL may be well tolerated while attenuating invasion into surrounding tissues and the metastatic spread of tumors.

For most of the pharmacological inhibitors tested in our study, concentrations suppressing 2D random cell migration also induce cytotoxicity. Hence, their effect on migration per se cannot be assessed. However, cytochalasins are an exception in this respect. Our findings indicate that cyB and cyD represent migrastatic compounds displaying a concentration window for the inhibition of cancer cell migration without cytotoxicity. Cytochalasins have previously been tested at high concentrations without assessing their toxicity. A concentration of 1 µM was reported to be sufficient to inhibit cellular growth [26]. Studies that tested the toxicity of cyD in a concentration range found toxic effects from 0.1 µM and higher in breast cancer cell lines [36]. Others observed that the highest non-toxic concentration of cyD was 0.01 to 0.1 µM for different cancer cell lines [23]. On the other hand, up to 50 µM cyD has been reported not to cause cytotoxicity in cancer and fibroblast cells [37]. Our findings reiterate that the impact of cyD on cell viability varies considerably between cell types, with 1.1 µM versus 41 nM as the highest tolerated dose in MV3 and Hs578T, respectively. For cyB, our findings concur with earlier reports showing that it is less toxic than cyD [38]. CyB has been less studied, but caused cell detachment, extracellular vesicle production and affected cell metabolism at higher concentrations [22,39,40].

The effect on the F-actin cytoskeleton of cytochalasins that we observe is in line with other reports [41,42]. Spheroids made of human colon cancer cell lines treated with 50 nM cyD also showed that actin filaments were disrupted, but the overall F-actin pattern marking the cell shape remained unchanged [26]. There is one other study describing the effect of 0.1 µM cyD on the 2D random cell migration of cancer cells, which also found that cell trajectories were reduced [23]. Likewise, 50 nM cyD blocked the migration of MV3 cells in a transmigration assay [24]. However, there is no published data currently available on the effect of cytochalasins on the migration of cancer cells from a tumor into the surrounding ECM. We show that concentrations of 750 nM cyB or 100 nM cyD suffice to suppress such behavior for Hs578T and MV3 cells. Importantly, our work allows us to discriminate such effects from any impact on cell viability, hence identifying these cytochalasins as migrastatic compounds. Others have shown that cytochalasins can also affect growth and EMT in tumor spheroids, but in these studies, migration into the surrounding ECM was not studied [25,26], and with the non-toxic concentrations used by us, we do not observe growth inhibition. Notably, the impact of cytochalasins on migration may involve overlapping as well as distinct mechanisms with the depletion of CFL. While the loss of CFL can lead to the stabilization of F-actin stress fibers at the expense of branching and network formation in protrusive structures [6,33], cytochalasins suppress F-actin fiber formation per se, thus also reducing stress fibers. However, cytochalasins not only act by direct interactions with F-actin, but cyD also interferes with CFL–actin interactions and CFL phosphorylation, thus providing an additional indirect role for cyD in the regulation of actin dynamics [17,18].

We find that both cyB and cyD strongly suppress cell stiffness at non-toxic concentrations. The short-term use of low µM concentrations of cyB or cyD did not affect cell stiffness as measured by AFM in other studies [20,43]. By contrast, Optics11 nanoindenter measurements did find a significant reduction in elastic moduli caused by 10 µM cyD, and a similar exposure led to reduced nuclear deformation on micropillars [36]. Our work differs from these studies as we test the longer-term exposure of verified non-toxic concentrations, and in this setting, we observe that stiffness is strongly reduced by cyB and cyD.

The softening effect of cytochalasins may explain their impact on 3D migration, but as discussed above, F-actin-mediated protrusions important for migration will also be affected. In agreement with the important role of optimal cortical stiffness, another study reduced cellular stiffness by depleting or inhibiting Rac1, which led to diminished 3D migration in ECM [44]. Moreover, increasing the cortical stiffness of MV3 cells by overexpressing NHE1, which activated CFL, led to enhanced invasion [24]. This suggests that, while invading cancer cells are typically softer than non-transformed cells [1,2,45,46] and the stiffness of multiple cancer types is inversely correlated with cell migration [47] through narrow gaps [48], either softening or stiffening these cells can lead to reduced migration efficacy as has been previously suggested by others [49]. Our study demonstrates for the first time that cytochalasins may be used to that effect, to suppress tumor cell migration at concentrations not affecting cell viability. A scenario would be to use such migrastatics to attenuate migration and invasion events in a primary tumor as well as metastatic lesions, in combination with tumor-targeted cytotoxic therapies, to improve patient outcome. This warrants further preclinical testing, where efficacy, but also toxicity due to effects on normal tissues, will have to be established using advanced culture models allowing prolonged exposure and ultimately in vivo models.

## 4. Materials and Methods

### 4.1. Cell Culture

The human breast cancer cell line Hs578T (ATCC, #HTB-126, Manassas, VA, USA), the human melanoma cell line MV3 [50], and hTERT-immortalized human fibroblasts (ATCC, #CRL-4001) were cultured in high-glucose Dulbecco’s Modified Eagle’s Medium (DMEM, Gibco, 11504496, Waltham, MA, USA) containing L-Glutamine and Sodium Pyruvate, supplemented with 10% fetal calf serum (Thermo Scientific, Waltham, MA, USA) and 25 µg/mL penicillin/streptomycin in a humidified incubator at 37 °C with 5% CO_2_.

### 4.2. Conditional Knockout Procedure

Dox-inducible Hs578T-Cas9 and MV3-Cas9 cell lines were generated using an Edit-R Inducible Lentiviral hEF1α-Blast-Cas9 Nuclease Plasmid DNA (Dharmacon, CAS11229, Lafayette, CO, USA). To do so, a subconfluent monolayer of Lenti-X 293T cells (Clontech, 632180, Mountain View, CA, USA) was transfected with a mixture of this plasmid and lentiviral helper vectors pMDLg-RRE, pCMV-VSVG, and pRSV-Rev (Addgene, #12251, #8454, and #12253, Watertown, MA, USA) using polyethylenimine (Polysciences, 23966-2, Warrington, PA, USA). After 24 h the medium was refreshed, and after 48 h and 72 h the medium was collected and filtered through a 0.45 µm filter. Next, Hs578T and MV3 cells were transduced with the virus-containing medium supplemented with 5 µg/mL Polybrene (Sigma-Aldrich, St. Louis, MO, USA). After 24 h the medium was changed and cells were selected using 2 µg/mL Blasticidin (R21001, Gibco, ThermoFisher Scientific, Waltham, MA, USA). Single clones of Hs578T-Cas9 and MV3-Cas9 were isolated, expanded, and tested for dox (Selleckchem, S5159, Houston, TX, USA)-inducible Cas9 expression. U6-gRNA/PGK-Puro-2A-BFP vectors expressing the short guide RNAs (sgRNAs) CCTCGTAGCAGTTTGCTTGCAAT or CCAGGGAGATGACGGCACTGCCC targeting CFL were obtained from the Sanger Whole Genome CRISPR Library (Sigma Aldrich). A non-targeting sgRNA was used as a negative control. Lentivirus production and the transduction of sgRNA vectors in the MV3-Cas9 and Hs578T-Cas9 cells were performed as described above for the Cas9 plasmid. Transduced cells were selected using puromycin (1 µg/mL for Hs578T and 16 µg/mL for MV3; A1113803, Gibco, ThermoFisher Scientific, Waltham, MA, USA). Target gene deletion was induced with 1 µg/mL dox for 4 days for Hs578T and 3 days for MV3. Cells were cultured for another 2–3 days without dox before conducting further experiments.

### 4.3. Pharmacological Inhibitors

CyB (C6762, Sigma-Aldrich, MO, USA), cyD (1233, Tocris Bioscience, Bristol, UK), Jasplakinolide (420107, Merck, Darmstadt, Germany), Blebbistatin Racemic (203389, CalBiochem, San Diego, CA, USA), Latruncilin A (3973, Tocris Bioscience, UK), GSK-269962 (4009, Tocris Bioscience, UK), and Bosutinib (SelleckChem, Huissen, The Netherlands) stocks were reconstituted and diluted in cell culture medium.

### 4.4. Proliferation Assay

A total of 2000 Hs578T and MV3 cells (control or knockout cells) were each seeded in a Cellstar 96-well plate (655180, Greiner, Alphen aan den Rijn, The Netherlands). Once attached, the cells were treated with concentration ranges of pharmacological inhibitors. Cell proliferation was assessed using the IncuCyte^®^ S3 (Sartorius, Goettingen, Germany), equipped with a 10× objective lens and placed inside an incubator at 37 °C and 5% CO_2_. Images were captured every 2 h over 3 days in the phase-contrast channel and confluency was assessed using the Incucyte software version 2020B (Essen BioScience Inc., Ann Arbor, MI, USA). The confluence was normalized to a time point of 0 h and was either plotted as a function of time or as a function of the inhibitor concentration, in which case the time point of 24 h was chosen.

### 4.5. F-Actin Morphology

Totals of 4000 Hs578T and 6000 MV3 cells (control or knockout cells) were seeded in a Screenstar 96-well plate (655866, Greiner, Alphen aan den Rijn, The Netherlands). Once attached, the cells were treated with concentration ranges of inhibitors. After 24 h, the cells were fixed with 2% formaldehyde (252549, Sigma-Aldrich, MO, USA) and stained with 0.1% Triton X-100 (T8787, Sigma-Aldrich, MO, USA), 0.4 µg/mL Hoechst 33258 (H21491, Invitrogen, Waltham, MA, USA) and 1:3000 AlexaFluor-488 Phalloidin (A12379, Thermo Fisher Scientific, MA, USA) or 0.06 µM Phalloidin Rhodamine (R415, Thermo Fisher Scientific, MA, USA) in PBS for 1 h at room temperature. F-actin morphology was analyzed by confocal microscopy (see below) and F-actin fibers versus ruffles were detected using ImageJ (version 1.53c). The mean stress fiber length was analyzed using the ridge detection plugin. The mean ruffle area was calculated using the analyze particles function.

### 4.6. Two-Dimensional Random Cell Migration

A Screenstar 96-well plate was coated with 20 µg/mL rat collagen type I, either isolated in-house (see Section 4.8, “Three-Dimensional ECM-Embedded Tumoroids”) or obtained commercially (R&D Systems, Minneapolis, MN, USA). Totals of 15,000 MV3 and 10,000 Hs578T cells per well were adhered overnight. The nuclei were stained with 1 µg/mL Hoechst 33258 for 1 h at 37 °C. The Hoechst 33258 was removed, and inhibitors and controls were added. As a positive migrastatic control, 1.25 µM Bosutinib was used, which has been shown to inhibit migration but not viability at the concentration used in our study [51]. Cells were imaged using the ImageXpress Micro Confocal High-Content Imaging System (Molecular Devices, San Jose, CA, USA, USA) equipped with the MetaXpress software Version 5.3.0.5 in an atmosphere of 37 °C with 5% CO_2_. Images were taken with a 20× objective lens over the course of 16 h, imaging 4 images per well every 12 min. The images were segmented using ImageJ (version 1.53c) with the Watershed Masked Clustering segmentation algorithm [52]. The tracking analysis was performed in CellProfiler (version 4.2.5) using the Overlap tracking method with a maximal pixel distance of 30. Tracking data from CellProfiler was imported into R (version 4.3.0) by using fixing track identifiers with the CPTrackR Shiny App [https://github.com/burgerga/CPTrackRApp/] developed in-house [53]. Speed was calculated using RStudio version 2023.06.1 Build 524.

### 4.7. Acoustic Force Spectroscopy (AFS)

Hs578T or MV3 cell suspensions were re-suspended in 20 µL of culture medium at an average concentration of 0.4 × 10^6^ cells/mL. In addition, 20 µL of 7.9 µm diameter silica beads (Spherotech, Inc., Lake Forest, IL, USA, SIP-60-10) were washed and re-suspended in 60 µL of culture medium. The beads were then mixed with the cell suspension at a cell-to-bead ratio of 1:1. The mixture was injected into the flow channel of the AFS microfluidic chip (Lumicks B.V., Amsterdam, The Netherlands) and incubated at 37 °C with 5% CO_2_. After 2 h of incubation, non-attached and dead cells were flushed out by connecting the AFS chip to a syringe pump (AL-2000, World Precision Instruments, Friedberg, Germany), and cells were cultured inside the incubator overnight at a slow flow rate (30 µL/min). After 24 h, the chip was placed on the AFS-G2 (Lumicks B.V., The Netherlands), which comprises a motorized z-stage mounted on an inverted microscope (Eclipse TE200, Nikon Instruments Inc., Melville, NY, USA), a G2 AFS chip holder, and a temperature controller. Visualization and illumination of the field of view were achieved by a motorized 20× objective microscope for a nanometer-precise z-translation, a red-light LED, and a uEye camera (UI-324xCP-M, IDS) at a sampling frequency of 60 Hz. Each bead inside the field of view was tracked by LabVIEW (provided by Lumicks B.V.), and their z-positions were determined by a look-up table set to track from 0 to 10,000 nm at a step size of 100 nm. A constant acoustic force was applied, pushing the bead upwards, thereby stretching the Hs578T cells. The standard linear liquid (SLL) model was used for data fitting and to quantify the viscoelastic properties according to the equations described in Evers et al. [54]. All measurements were performed at 37 °C with a peak-to-peak driving voltage of 60 Vpp at 14.51 MHz frequency. To probe the effects of inhibitors on cell mechanics, the AFS chip was placed back inside the incubator after baseline measurements and reconnected to the syringe pump for exposure of cells to culture medium supplemented with inhibitors for 24 h.

### 4.8. Three-Dimensional ECM-Embedded Tumoroids

A collagen type I solution was isolated from rat-tail collagen by acid extraction as described previously [55]. We created 2 mg/mL collagen matrices by mixing it with DMEM (Gibco, ThermoFisher Scientific, Waltham, MA, USA), HEPES 0.1 M (1 M stock, Biosolve, Valkenswaard, The Netherlands), and NaHCO_3_ 44 mM (Merck, Darmstadt, Germany) as previously described [56]. A 30 µL collagen mixture was polymerized in a 384-well CELLSTAR^®^ plate (781091, Greiner Bio-one, Kremsmünster, Austria) at 37 °C for 1 h. ECM-embedded tumoroids were generated as described previously [57]. In short, tumor cell suspensions (control or knockout cells) in PBS containing 2% polyvinylpyrrolidone (PVP, P5288, Sigma-Aldrich) were made containing ~50.000 cells per µL and printed into collagen matrices at defined x-y-z positions 150 μm above the bottom of the wells using the injection robotics from Life Science Methods (Leiden, The Netherlands). The initial cell number of the tumoroids was ~2500 and the diameter was 200 µm. Tumoroids were incubated with a concentration range of inhibitors at 37 °C. After 24 h tumoroids were stained with 2 µg/mL Hoechst 33258 and 0.4 µM Propidium Iodide (PI, 324811, BD, Franklin Lakes, NJ, USA) for 1 h, followed by live confocal microscopy. Tumoroids were subsequently fixed with 2% formaldehyde and stained with 0.1% TritonX-100, 0.4 µg/mL Hoechst 33258, and 1:3000 AlexaFluor-488 Phalloidin (Thermo Fisher, A12379) or 0.06 µM Phalloidin Rhodamine in PBS for 3–4 h at room temperature and analyzed by confocal microscopy. Z-stacks of confocal images of tumoroids were projected using OMERO.web (version 5.22.1). Scanning confocal z-stacks of the actin cytoskeleton were projected using the standard deviation in the z-direction. A Gaussian filter with a narrow kernel was used to remove small fluctuations from the projected image. An adaptive threshold was used to separate the foreground from the background. The area of the tumoroids was calculated using the generated foreground image. To analyze 3D migration from the tumoroids into the ECM scaffolds, a home-built analysis tool in Matlab version 2020b was used [58].

### 4.9. Scanning Confocal Microscopy

Images of 2D plated cells for morphology analysis were acquired on an Eclipse Ti inverted scanning confocal microscope equipped with an automated stage, four laser lines at 405 nm, 488 nm, 561 nm and 640 nm, an A1R MP scanner, and a Plan Fluor ×40/0.75 NA objective (Nikon Instruments Inc., Melville, NY, USA). The camera was controlled through NIS Element software version 6.10 (Nikon Instruments Inc., Melville, NY, USA). Scanning confocal fluorescence microscopy of the ECM-embedded tumoroids was performed on the same microscope using a Plan Apo ×20/0.75 NA objective lens with a 20 µm distance between Z-slices. Images of tumoroids on day 0 were acquired with Hoechst staining using the 405 nm laser line and transmission detection channel. Images of tumoroids on day 2 were additionally acquired with the laser line at 561 nm, to detect PI. Images of fixed tumoroids were additionally acquired with the laser lines at 561 nm or 488 nm for the detection of phalloidin (F-actin marked red for MV3 and green for Hs578T throughout this paper).

### 4.10. Data Visualization and Statistical Analysis

For data visualization and statistical analysis, GraphPad Prism (version 8.1.1) was used. For the presentation of images and graphs, Adobe Illustrator (version 27.9) was used. The Area Under the Curve (AUC) was calculated for the proliferation graphs. The Shapiro–Wilk test was used to test for normality. When datasets showed normal distribution, an unpaired, parametric two-tailed *t*-test was used. When datasets were significantly different from normal distribution, an unpaired, non-parametric two-tailed *t*-test (Mann–Whitney) was applied. Means of biological replicates were used for all bar graphs or AUC data (n = 2 or n = 3) to determine statistical significance between populations. Datasets were considered significantly different with probabilities of *p* < 0.0001 (****), *p* < 0.001 (***), *p* < 0.01 (**), and *p* < 0.05 (*), and not significantly different for probabilities of *p* > 0.05 (ns).

## Figures and Tables

**Figure 1 ijms-26-07021-f001:**
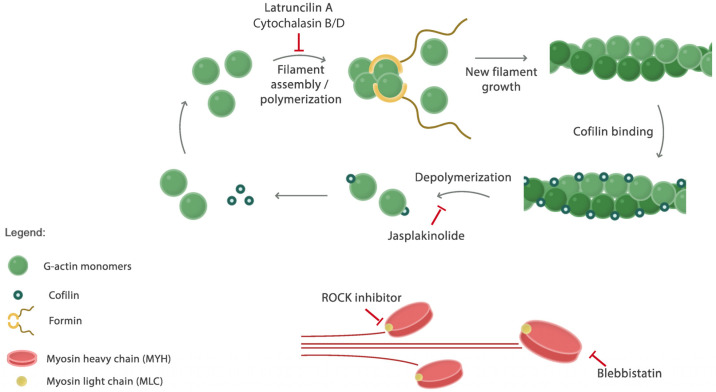
A schematic representation of actin filament turnover. A cartoon illustrating the dynamic processes involved in actin cytoskeleton remodeling. Actin monomers (G-actin) polymerize to form filaments (F-actin), a process inhibited by Latruncilin A and cytochalasins. Formin proteins facilitate the elongation of filament growth. Severing proteins such as cofilin enhance turnover by fragmenting filaments, increasing the number of free ends at which depolymerization occurs. Depolymerization occurs predominantly at the pointed end and can be inhibited by Jasplakinolide. Cytoskeletal contractility is generated by myosin motors that are inhibited by blebbistatin and are under the control of ROCK-mediated myosin light-chain phosphorylation.

**Figure 2 ijms-26-07021-f002:**
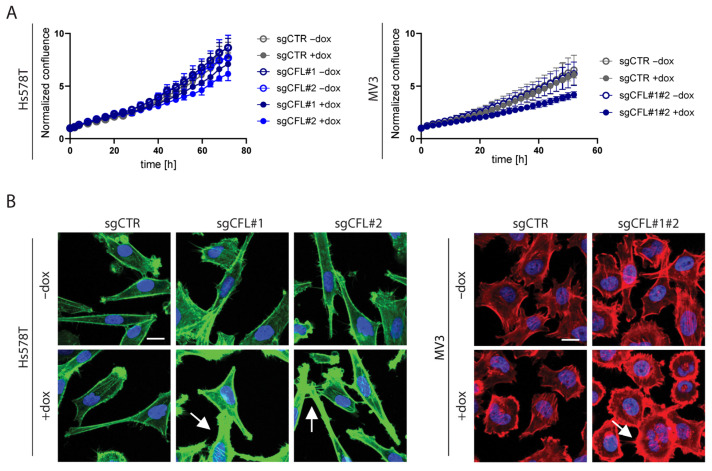
The effect of CFL knockout on proliferation and F-actin morphology. (**A**): The effect of CFL knockout on cell proliferation, analyzed with IncuCyte. Confluence is normalized to time point 0 h and plotted as a function of time (mean ± SEM). N = 2 biological replicates, each with 10 technical replicates. (**B**): Representative confocal images of cells seeded in 2D and stained with Hoechst (nucleus, blue) and Phalloidin (F-actin, green for Hs578T and red for MV3). The arrows indicate the increased accumulation of F-actin at the cell cortex in the CFL knockout cells. Scalebar = 20 µm. One of two biological repeats, each with at least three technical replicates, is shown.

**Figure 3 ijms-26-07021-f003:**
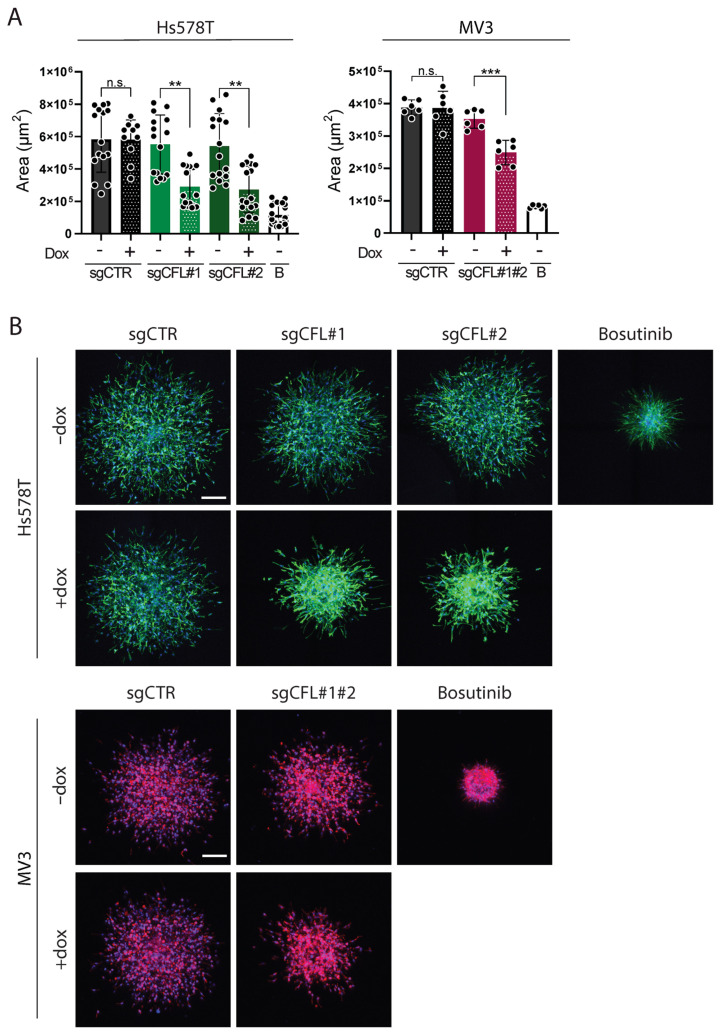
The effect of CFL knockout on the migration of 3D ECM-embedded tumoroids. (**A**): The area quantifications of the z-projections of tumoroids 24 h post-injection. The mean ± SD of all tumoroids that were analyzed is shown for 2–3 biological replicates, each containing a minimum of 3 tumoroids. The protein tyrosine kinase inhibitor bosutinib (B) was used as a positive control, which suppressed 2D cell migration without affecting growth at a concentration of 1.25 µM. n.s, non-significant; ** *p* < 0.01; *** *p* < 0.001. (**B**): Representative confocal images of the z-projection of tumoroids with CFL knockouts 24 h post-injection. The tumoroids were stained with Hoechst (nucleus, blue) and Phalloidin (F-actin, green for Hs578T and red for MV3). The positive migrastatic control was 1.25 µM bosutinib. Scalebar = 200 µm.

**Figure 4 ijms-26-07021-f004:**
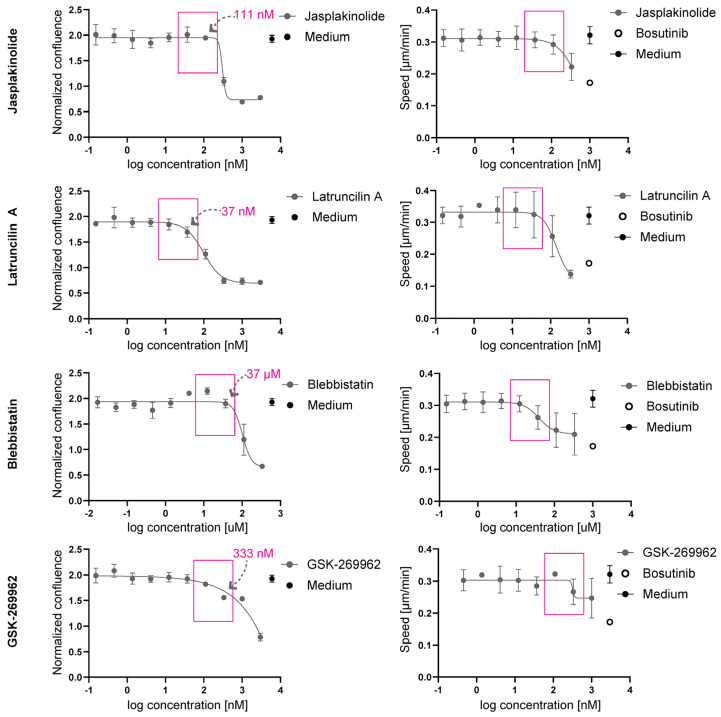
The effects of Jasplakinolide, Latruncilin A, Blebbistatin, and GSK-269962 on 2D random migration. (**Left panels**): The proliferation curves of Hs578T cells seeded in 2D, showing confluence after 24 h of exposure to concentration ranges of the indicated inhibitors, normalized to time point 0. The boxes indicate the highest concentrations tolerated without growth suppression. The mean ± SEM is shown for N = 3 biological replicates, each containing 10 technical replicates. (**Right panels**): The random cell migration speed of subconfluent Hs578T cells seeded in 2D and exposed to concentration ranges of inhibitors. The boxes indicate the same maximum tolerated concentrations boxed as in the (**left panels**). Bosutinib was used as a positive control. The mean ± SEM is shown for 2 biological replicates, each containing 4 technical replicates.

**Figure 5 ijms-26-07021-f005:**
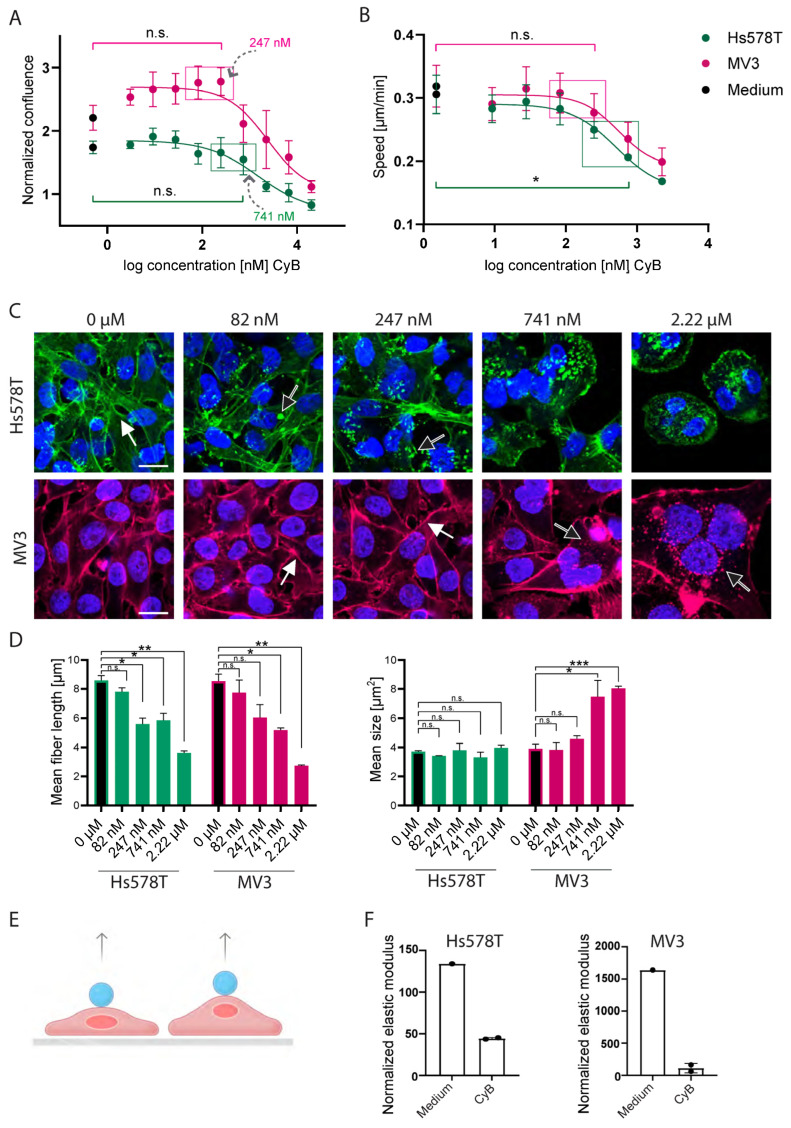
CyB suppresses 2D migration speed and cell stiffness at non-toxic concentrations, without disrupting F-actin. (**A**): The proliferation curves of Hs578T and MV3 cells seeded in 2D, showing confluence after 24 h of exposure to concentration ranges of the indicated inhibitors, normalized to time point 0. The boxes indicate the highest concentrations tolerated without growth suppression. The mean ± SEM is shown for N = 3 biological replicates, each with 10 technical replicates. n.s., non-significant. (**B**): The random cell migration speed of subconfluent Hs578T and MV3 cells seeded in 2D and exposed to concentration ranges of cyB. The boxes indicate the same maximum tolerated concentrations boxed in the left panel. The mean ± SEM is shown for N = 3 biological replicates, each with 8 technical replicates. n.s., non-significant; * *p* < 0.05. (**C**): F-actin fibers (Phalloidin, green or red) and nuclei (Hoechst, blue) of Hs578T and MV3 cells seeded in 2D in the absence or presence of the indicted cyB concentrations. The black arrows indicate actin aggregates and structures resembling membrane ruffles appearing with increasing inhibitor concentrations and replacing the F-actin stress fibers observed under control conditions (white arrows). Scalebar = 20 µm. (**D**): The stress fiber length (left; indicated in (**C**) with white arrows) and area of membrane ruffles (right; indicated in (**C**) with black arrows). The mean ± SD for one experiment with two replicates is shown. n.s., non-significant; * *p* < 0.05; ** *p* < 0.01; *** *p* < 0.0001. (**E**): A cartoon illustrating the principle of AFS. Acoustic force pushes the bead upwards, thereby stretching the cells, with the amount of stretch inversely related to stiffness. (**F**): AFS measurements for cells exposed for 24 h to a control medium or cyB (741 nM for Hs578T and 247 nM for MV3), normalized to the measurement before exposure. The mean ± SD for one experiment of two, each containing 2 technical replicates, is shown.

**Figure 6 ijms-26-07021-f006:**
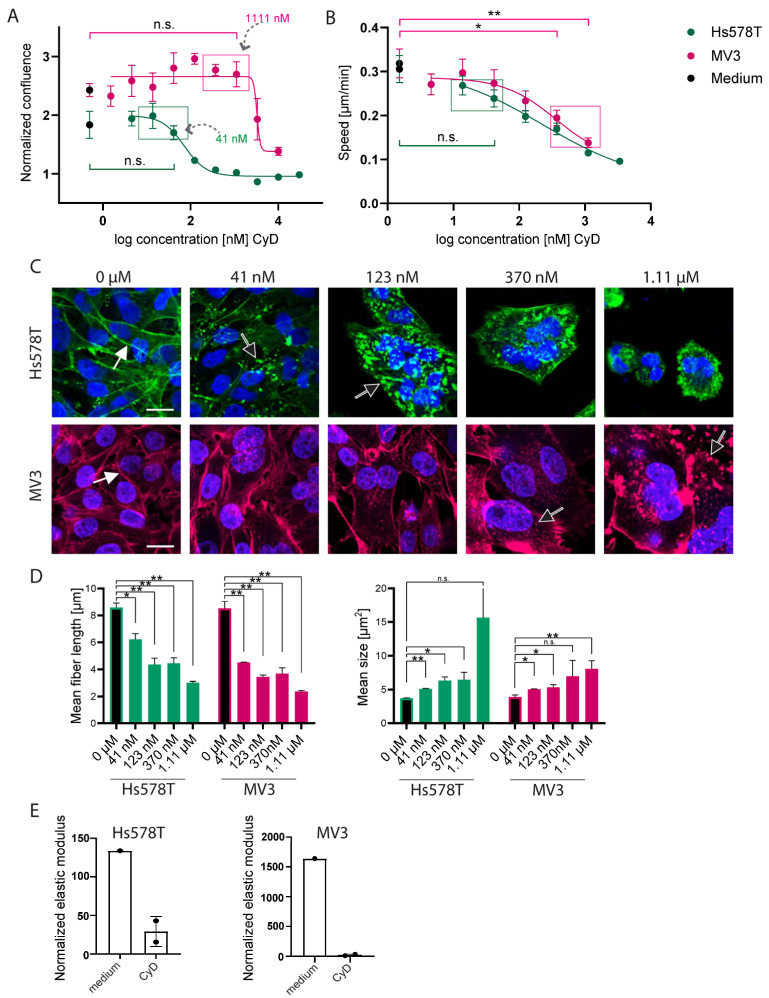
CyD suppresses 2D migration speed and cell stiffness at non-toxic concentrations, without disrupting F-actin. (**A**): Proliferation curves of Hs578T and MV3 cells seeded in 2D showing confluence after 24 h of exposure to concentration ranges of the indicated inhibitors, normalized to time point 0. The boxes indicate the highest concentrations tolerated without growth suppression. The mean ± SEM is shown for N = 3 biological replicates, each with 10 technical replicates. (**B**): The random cell migration speed of subconfluent Hs578T and MV3 cells seeded in 2D and exposed to concentration ranges of cyD. The boxes indicate the same maximum tolerated concentrations boxed in the left panel. The mean ± SEM is shown for N = 3 biological replicates, each with 8 technical replicates. n.s., non-significant; * *p* < 0.05; ** *p* < 0.01. (**C**): F-actin fibers (Phalloidin, green or red) and nuclei (Hoechst, blue) of Hs578T and MV3 cells seeded in 2D in the absence or presence of the indicted cyD concentrations. The black arrows indicate actin aggregates and structures resembling membrane ruffles appearing with increasing inhibitor concentrations and replacing the F-actin stress fibers observed under control conditions (white arrows). Scalebar = 20 µm. (**D**): the Stress fiber length (left; indicated in (**C**) with white arrows) and area of membrane ruffles (right; indicated in (**C**) with black arrows). The mean ± SD for one experiment with two replicates is shown. n.s, non-significant; * *p* < 0.05; ** *p* < 0.01. (**E**): AFS measurements for cells exposed for 24 h to a control medium or cyD (41.2 nM for Hs578T and 123 nM for MV3), normalized to the measurement before exposure. The mean ± SD for one experiment of two, each containing 2 technical replicates, is shown.

**Figure 7 ijms-26-07021-f007:**
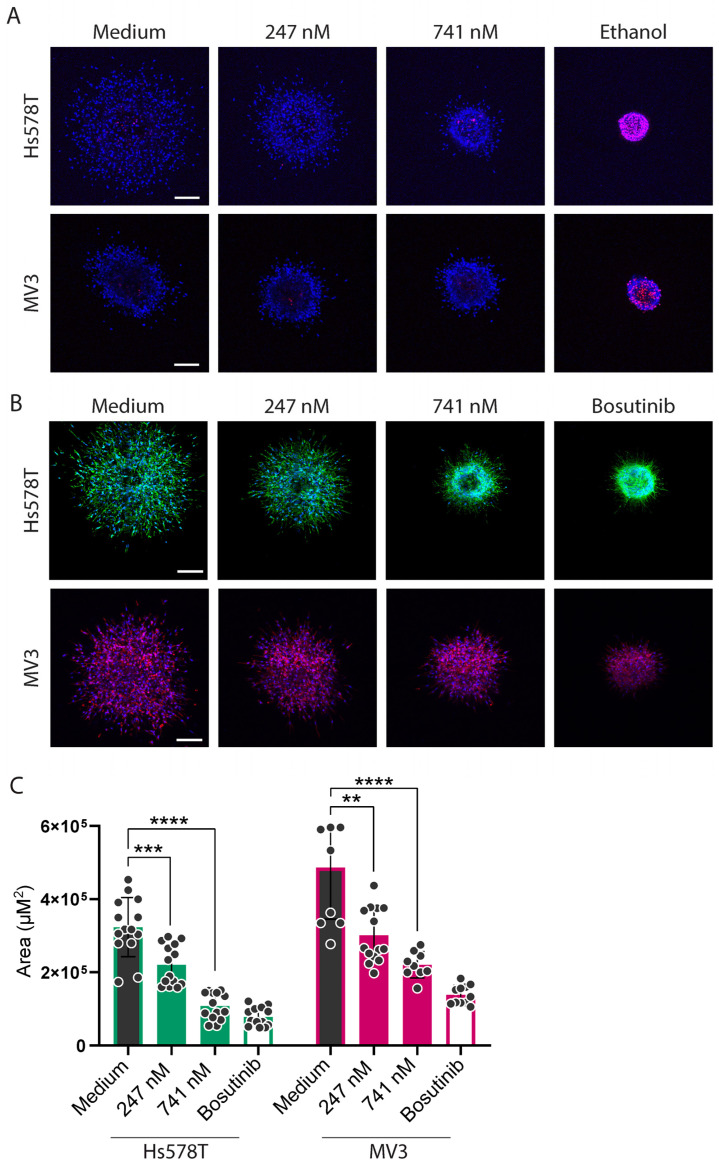
CyB suppresses 3D migration from tumoroids into the surrounding ECM at non-toxic concentrations. (**A**): Representative confocal images of live 3D ECM-embedded tumoroids derived from the indicated cell lines and treated with the indicated concentrations of cyB, stained with Hoechst (nucleus, blue) and Propidium Iodide (PI, dead cells, red). The images were taken at 24 h post-injection of tumor cell clusters into the ECM. Ethanol was used as a positive PI control. Scalebar = 200 µm. (**B**): Representative confocal images of fixed ECM-embedded tumoroids derived from the indicated cell lines and treated with the indicated concentrations of cyB, stained with Hoechst (nucleus, blue) and Phalloidin (F-actin, green for Hs578T and red for MV3). The images were taken at 24 h post-injection of tumor cell clusters into the ECM. Bosutinib was used as a positive control. Scalebar = 200 µm. One out of three biological replicates are shown. (**C**): The area quantifications of the z-projection of the tumoroids. The mean ± SD of all the tumoroids that were analyzed is shown for three biological replicates with at least 4 tumoroids each. ** *p* < 0.01; *** *p* < 0.001; **** *p* < 0.0001.

**Figure 8 ijms-26-07021-f008:**
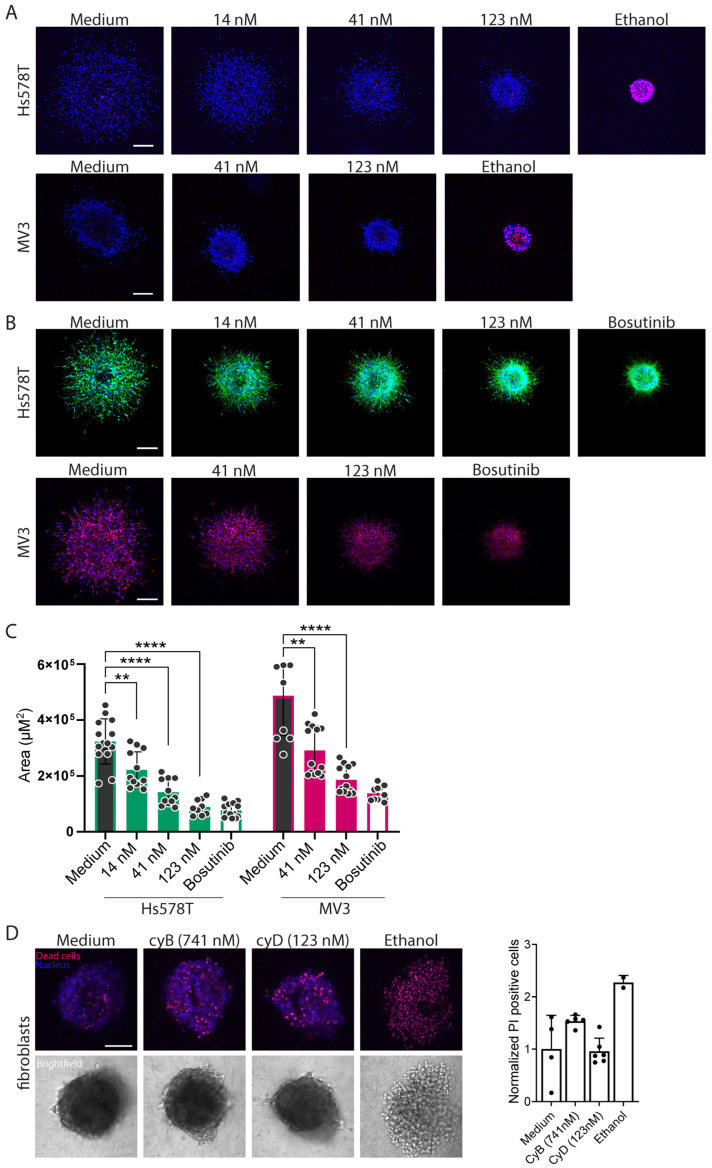
CyD suppresses 3D migration from tumoroids into the surrounding ECM at non-toxic concentrations. (**A**): Representative confocal images of live 3D ECM-embedded tumoroids derived from the indicated cell lines and treated with the indicated concentrations of cyD, stained with Hoechst (nucleus, blue) and Propidium Iodide (PI, dead cells, red). The images were taken at 24 h post-injection of tumor cell clusters into the ECM. Ethanol was used as a positive PI control. Scalebar = 200 µm. (**B**): Representative confocal images of fixed ECM-embedded tumoroids derived from the indicated cell lines and treated with the indicated concentrations of cyD, stained with Hoechst (nucleus, blue) and Phalloidin (F-actin, green for Hs578T and red for MV3). Images taken at 24 h post-injection of tumor cell clusters into the ECM. Bosutinib was used as a positive control. Scalebar = 200 µm. One out of three biological replicates are shown. (**C**): The area quantifications of the z-projection of the tumoroids. The mean ± SD of all the tumoroids that were analyzed is shown for three biological replicates with at least 4 tumoroids each. ** *p* < 0.01; **** *p* < 0.0001. (**D**): Representative confocal images (left) and the quantification (right) of live 3D ECM-embedded tumoroids derived from hTERT-immortalized human fibroblasts and treated with the indicated concentrations of cyB or cyD, stained with Hoechst (nucleus, blue) and Propidium Iodide (PI, dead cells, red). The images were taken at 24 h post-injection of fibroblast clusters into the ECM. Ethanol was used as a positive PI control. Scalebar = 100 µm. One biological replicate with 2–4 technical replicates is shown.

## Data Availability

Data is contained within the article.

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
