# Peer review of "Cytochalasins Suppress 3D Migration of ECM-Embedded Tumoroids at Non-Toxic Concentrations"

_ijms, 2025, doi:10.3390/ijms26147021_

Round 1
Reviewer 1 Report
Comments and Suggestions for Authors
This manuscript explores an important area of migrastatic strategies by targeting cytoskeletal remodeling in 3D tumoroid models. The combined use of CFL knockout and cytochalasin treatment is well justified and of translational relevance. However, the interpretation that decreased cellular stiffness alone accounts for migration inhibition appears to be an oversimplification and would benefit from further clarification and supporting data.
-
The authors propose that reduced stiffness leads directly to impaired migration, but this relationship is not fully substantiated. Several studies have reported increased migratory capacity in softer cells, suggesting that stiffness alone may not be the determining factor. Additional evidence or discussion is needed to justify this mechanistic conclusion.
-
To more conclusively determine whether migration inhibition results specifically from changes in stiffness rather than other structural alterations, the following experiments are recommended:
2-1. Use pharmacological agents (e.g., ROCK inhibitors such as Y-27632) that modulate cellular stiffness without directly disrupting F-actin polymerization to assess effects on migration.
2-2. Quantitatively analyze actin structures (lamellipodia, filopodia) and focal adhesion complexes to differentiate between mechanical and architectural influences on migration.
2-3. If feasible, traction force microscopy could provide direct measurements of migratory force generation post-treatment. -
Although both CFL knockout and cytochalasin treatment reduce migration, their underlying mechanisms differ fundamentally. CFL knockout impairs actin turnover and promotes stress fiber accumulation, whereas cytochalasin inhibits actin polymerization itself. Treating these interventions as mechanistically equivalent without additional validation weakens the interpretation. Experimental data or literature support clarifying whether migration inhibition arises from convergent or distinct pathways would strengthen the manuscript.
-
The manuscript integrates 2D migration assays and 3D ECM invasion results without adequately addressing the well-known differences in cellular behavior between these environments. Given the substantial variation in stiffness, adhesion dynamics, and matrix architecture between 2D and 3D contexts, more discussion or experiments supporting the direct correlation of 2D migration inhibition with 3D invasion suppression are warranted.
-
The claim that cytochalasin inhibits migration at non-toxic concentrations requires clearer definition and validation. The observed polyploidy and multinucleation suggest cytoskeletal disruptions potentially affecting cell division. A thorough assessment of cytotoxicity—including extended viability assays and analyses of cell cycle progression and nuclear morphology—would substantiate the assertion of non-toxicity.
Author Response
Reviewer 1.
This manuscript explores an important area of migrastatic strategies by targeting cytoskeletal remodeling in 3D tumoroid models. The combined use of CFL knockout and cytochalasin treatment is well justified and of translational relevance. However, the interpretation that decreased cellular stiffness alone accounts for migration inhibition appears to be an oversimplification and would benefit from further clarification and supporting data.
- The authors propose that reduced stiffness leads directly to impaired migration, but this relationship is not fully substantiated. Several studies have reported increased migratory capacity in softer cells, suggesting that stiffness alone may not be the determining factor. Additional evidence or discussion is needed to justify this mechanistic conclusion. We fuly agree with the comment that the genetic and pharmacological experiments performed in this paper will influence a range of F-actin depedent processes, incuding cortical stiffness and actin-based protrusions involved in migration. In fact, we do not conclude in this manuscript that we demonstrate reduced stiffness as the sole mechanism underlying the migrastatic effects observed. In the abstract and introduction we clearly point out the broad range of actin-dependent processes driving migration. We discuss that these will all be affected by our experimental approaches. Our conclusions are limited to the novel key finding that migrastatic effects can be distinguished from effects on viability, which is not trivial. To make this point more clear in the last paragraph of the discussion, where we indicate the connection between reduced stiffness and reduced migration, we have now added more explicitely the statement that effects on pro-migratory membrane protrusions may certainly be involved as well.
- To more conclusively determine whether migration inhibition results specifically from changes in stiffness rather than other structural alterations, the following experiments are recommended:
2-1. Use pharmacological agents (e.g., ROCK inhibitors such as Y-27632) that modulate cellular stiffness without directly disrupting F-actin polymerization to assess effects on migration. We used the GSK 269962 ROCK inhibitor and show that this compound lacks a good window to discriminate its migrastatic effect from its cytotoxic effect (figure 4).
2-2. Quantitatively analyze actin structures (lamellipodia, filopodia) and focal adhesion complexes to differentiate between mechanical and architectural influences on migration. The experiments we performed do not allow us to discriminate such structures acurately (other than an assumption based on shape). We do however, fully agree with the reviewer that a quantification of the observed phenotypic shift from a contractile phenotype with clear stress fibers to a phenotype where stress fibers disapear and rufles become visible is important. We have performed this analysis, the data is shown in revised figures 5 and 6. - 2-3. If feasible, traction force microscopy could provide direct measurements of migratory force generation post-treatment. We agree, this would be a very good addition. Unfortunately we do not have access to traction force microscopy.
- Although both CFL knockout and cytochalasin treatment reduce migration, their underlying mechanisms differ fundamentally. CFL knockout impairs actin turnover and promotes stress fiber accumulation, whereas cytochalasin inhibits actin polymerization itself. Treating these interventions as mechanistically equivalent without additional validation weakens the interpretation. Experimental data or literature support clarifying whether migration inhibition arises from convergent or distinct pathways would strengthen the manuscript. We agree with the reviewer that the impact of CFL depletion and cytochalasin treatment, while both reducing migration, may involve overlapping but also distinct mechnisms. We describe in the introduction chapter the role of CFL in F-actin dynamics and the mechanism of action of the inhibitors used, including the cytochalasins. We also describe there, the literature showing how cytochalasins act through direct interaction with F-actin but can also interfere with CFL phosphorylation and CFL-actin interactions, thus indirectly disturbing actin dynamics through CFL. Such aspects were also mentioned in the discussion chapter. In this revision, we have added these considerations and a direct comparison between the two strategies more prominantly in the discussion.
- The manuscript integrates 2D migration assays and 3D ECM invasion results without adequately addressing the well-known differences in cellular behavior between these environments. Given the substantial variation in stiffness, adhesion dynamics, and matrix architecture between 2D and 3D contexts, more discussion or experiments supporting the direct correlation of 2D migration inhibition with 3D invasion suppression are warranted. We fully agree that 2D migration and 3D migration cannot be directly compared. Moving on a 2D surface versus navigating a 3D ECM network has been shown by many studies, including our own, to involve distinct mechanims. For the CFL KO studies, we did not include 2D. This has been done by others. Our finding that the CFL KO prevents 3D migration of cancer cells, while to be expected, is novel. We did use the 2D migration experiments to identify pharmacological inhibitor concentrations that impacted migration but not cytotoxicity. In fact, these concentrations selected in 2D translated quite well to - and actually provided a stronger migrastatic effect in - the 3D environment. In this revision, we have added the rationale for this approach now more explictely in the abstract and the results section.
- The claim that cytochalasin inhibits migration at non-toxic concentrations requires clearer definition and validation. The observed polyploidy and multinucleation suggest cytoskeletal disruptions potentially affecting cell division. A thorough assessment of cytotoxicity—including extended viability assays and analyses of cell cycle progression and nuclear morphology—would substantiate the assertion of non-toxicity. With respect to polypoidy/ multinucleation we describe in the results “Polyploid cells appeared at high concentrations of cyB for both cell lines and MV3 showed extremely large flattened multinucleated cells at 2 µM.” Importantly we do not use those high concentrations in any of the subsequent experiments where we determine that cytochalasins (cyB and cyD) can inhibit migration at non-toxic concentrations. Instead, the purpose of this dose range is that we can use concentrations far below such toxic phenomena, selecting a range where 2D growth is not affected. Moreover, for those selected tollerated concentrations we show that while migration is severly impaired in 3D, tumor growth is not affected and no signs of loss of viability in 3D are observed (indicated by an absence of PI staining). Therefore, we believe this warrants the conclusion that cyB and cyD can inhibit migration of tumor cells in 3D ECM at non-toxic concentrations. In this revision, we have extended these observations to human fibroblasts (see below).
Reviewer 2 Report
Comments and Suggestions for Authors
The authors address a crucial issue in the fight against cancer: its ability to spread throughout the body. While studies have so far been conducted in vitro, are the authors considering in vivo studies using cytB and cytD? Would the cytB and cytD solutions used be toxic to the physiologically normal cells surrounding the tumour? As these are compounds that block cell division, could they damage cells in a particular organ or tissue when administered to a tumour?
In the literature references, the authors usually begin the citation with the author's name, but in a few cases the author's initials appear instead. I suggest making all items in the index consistent.
Author Response
Reviewer 2.
The authors address a crucial issue in the fight against cancer: its ability to spread throughout the body. While studies have so far been conducted in vitro, are the authors considering in vivo studies using cytB and cytD? Would the cytB and cytD solutions used be toxic to the physiologically normal cells surrounding the tumour? As these are compounds that block cell division, could they damage cells in a particular organ or tissue when administered to a tumour? The reviewer addresses an important point for follow up studies. We have included in this revision a first test, showing that non-cancerous cells, i.e. human fibroblasts, tollerate the selected migrastatic concentrations at least in the 24h exposure assays that we have used in this study. The new data is included in figure 8 and described at the end of the results section. It is still certainly possible that toxicity in surrounding normal tissue limits or even precludes the use of these compounds especially upon prolonged exposure. We have indicated this issue in the revision at the end of our discussion.
In the literature references, the authors usually begin the citation with the author's name, but in a few cases the author's initials appear instead. I suggest making all items in the index consistent. We have harmonized the references as requested.
Reviewer 3 Report
Comments and Suggestions for Authors
The strengths of this study are as follows:
- Evaluation of Effects at Non-Toxic Concentrations ​
The study clearly demonstrates the migrastatic effects of Cytochalasin B (CyB) and Cytochalasin D (CyD) at non-toxic concentrations. While previous studies have reported toxicity at higher concentrations, this research highlights the potential to suppress cell migration while maintaining cell viability, increasing the likelihood of clinical application. ​
- Assessment in 3D Environments
Unlike studies limited to 2D cultures, this research evaluates the effects of CyB and CyD in 3D ECM (extracellular matrix)-embedded tumor models. This approach provides a more biologically relevant environment, enhancing the reliability of the findings.
- Impact on Cell Stiffness ​
The study shows that Cytochalasins reduce cell stiffness and suggests that this softening contributes to migration suppression. Evaluating biomechanical properties like cell stiffness is a unique aspect not commonly addressed in other studies. ​
- Combination of Genetic and Pharmacological Approaches ​
In addition to pharmacological methods, the study uses conditional knockout models for cofilin (CFL) to investigate the genetic suppression of cell migration. This comprehensive approach allows for a deeper understanding of the interaction between genetic factors and drug effects. ​
- Multifaceted Experimental Methods
The study evaluates drug effects from multiple perspectives, including cell proliferation, migration speed, cell stiffness, and F-actin morphology changes. This multidimensional analysis provides a detailed understanding of the mechanisms of action. ​
- Long-Term Effects at Non-Toxic Concentrations ​
The research assesses the long-term effects of Cytochalasins at verified non-toxic concentrations, addressing the limitations of previous studies that focused on short-term toxicity. This highlights the drugs' safety and sustained efficacy. ​
- Proposal of Novel Migrastatic Drug Candidates ​
CyB and CyD are proposed as "migrastatic drugs," offering a solution to the toxicity and side effects associated with existing drugs. This positions them as promising candidates for suppressing cancer cell migration. ​
These strengths make this study stand out compared to previous research, as it evaluates drug safety and efficacy comprehensively and under biologically relevant conditions.
​
This research describes detailed experimental results and methodology, and meets the basic requirements of an academic paper, but may be lacking in the following areas:
- Reference to clinical applications
The effectiveness of CyB and CyD at non-toxic concentrations has been shown, but there is a lack of discussion about the possibility and challenges of these drugs moving forward to clinical trials. A concrete outlook for clinical applications would make the practicality of the research clearer.
- Evaluation of long-term effects
The short-term effects of the drugs have been evaluated, but there is a lack of data on the effects of long-term use on cells and tissues. This makes it difficult to evaluate the safety and sustained effects of the drugs.
- Comparison with other drugs
Other migratory inhibitors (e.g., Jasplakinolide, Latrunculin A, Blebbistatin, etc.) are mentioned, but direct comparisons with those drugs are lacking. Comparative studies would make the superiority of Cys clearer.
- Use of diverse cell models
The study mainly focuses on Hs578T and MV3 cells, but verifying the effects in other types of cancer cells and normal cells will improve the generalizability of the results.
- Detailed elucidation of the mechanism
Although the effects of Cys on cell stiffness and migration have been described, detailed elucidation of the mechanism of action at the molecular level is lacking. This will allow for a clearer targeting of the drug.
- Statistical details
Statistical methods are briefly mentioned, but details on the distribution of data and testing methods may be lacking. This may reduce the reliability of the results.
Adding these points would further improve the scientific depth and practical applicability of this paper.

Author Response
Reviewer 3.
Reference to clinical applications
The effectiveness of CyB and CyD at non-toxic concentrations has been shown, but there is a lack of discussion about the possibility and challenges of these drugs moving forward to clinical trials. A concrete outlook for clinical applications would make the practicality of the research clearer. We have added such outlook at the end of the discussion chapter. Notably, we have kept this very short since this is speculative. Our findings for the first time show that cytochalasins can in fact be used as migrastatics at low non-toxic concentrations in the context of 3D cancer migration, but translation to clinical application requires the initiation of a series of much more extensive preclinical studies.
Evaluation of long-term effects
The short-term effects of the drugs have been evaluated, but there is a lack of data on the effects of long-term use on cells and tissues. This makes it difficult to evaluate the safety and sustained effects of the drugs. We agree with the reviewer, as also answered to reviewer 2, that while our data are novel (and we believe make an important contribution to the field where these inhibitors are often used without controlling toxicity), they are nevertheless a first step. We have not addressed effects in more elaborate precinical models, where tissues are exposed for long periods. Our finding that the cytochalasins can be used as migrastatics is novel. However, whether they can be used long term and might be safe in vivo is not addressed by us.
Comparison with other drugs
Other migratory inhibitors (e.g., Jasplakinolide, Latrunculin A, Blebbistatin, etc.) are mentioned, but direct comparisons with those drugs are lacking. Comparative studies would make the superiority of Cys clearer. The goal for the pharmacological inhibitors tested by us was to select only those compounds with a possible migrastatic action. We have moved a small panel of inhibitors through the first testing phase described in figure 4-6. All inhibitors were compared at this first stage. The inhibitors in figure 4, refered to by the reviewer, were not selected for further analysis based on their inability to suppress migration at below-cytotoxic concentrations, i.e., they were not migrastatic.
Use of diverse cell models and detailed elucidation of the mechanism
The study mainly focuses on Hs578T and MV3 cells, but verifying the effects in other types of cancer cells and normal cells will improve the generalizability of the results. Although the effects of Cys on cell stiffness and migration have been described, detailed elucidation of the mechanism of action at the molecular level is lacking. This will allow for a clearer targeting of the drug. Indeed, we show in this study that in two cell lines representing two different cancer types, the same two cytochalasins are selected as migrastatic. In this revision we have added non-cancerous fibroblasts to further show that the compounds are not toxic at the identified concentrations in the tissue culture setting. Further characterization in a wider panel of cell types and further investigation of the MoA is indeed of interest for a follow up study.
Statistical details
Statistical methods are briefly mentioned, but details on the distribution of data and testing methods may be lacking. This may reduce the reliability of the results. The description of the statistical methods has been extended as suggested by the reviewer. In fact, the reviewer has pointed out an error in our image analysis: we have observed that a normal distribution cannot be applied to the 3D migration data. We have therefore reanalyzed this data using non-normal distribution. Figures 7 and 8 and the description of the statistical methods have been revised accordingly.
Round 2
Reviewer 1 Report
Comments and Suggestions for Authors
These revisions substantially enhance the manuscript’s mechanistic rigor and interpretive clarity. I believe the study’s core contributions distinguishing migrastatic effects from viability effects and demonstrating their relevance in a 3D setting are now presented with the depth and precision they deserve.
Author Response
We sincerely thank Reviewer 1 for their thoughtful feedback and generous assessment of our revised manuscript. We deeply appreciate your recognition of the enhanced mechanistic rigor and interpretive clarity, as well as your acknowledgement of our efforts to distinguish migrastatic effects from viability effects in a 3D context.
Reviewer 3 Report
Comments and Suggestions for Authors
Thank you for providing answers to my questions and doing the corrective actions accordingly.
I appreciate the effort of the authors.
I have read in detail the responses and they address the issues raised.
It appears much stronger and suitable for publication.
Author Response
We are truly grateful to Reviewer 3 for their supportive comments and careful evaluation of our revisions. Thank you for recognizing the efforts we made in addressing your questions and concerns. Your positive feedback on the strengthened presentation and improved suitability for publication is highly encouraging and appreciated.